# High-resolution characterization of gene function using single-cell CRISPR tiling screen

Lu Yang[1,7], Anthony K. N. Chan[1,7], Kazuya Miyashita[1,7], Christopher D. Delaney[2], Xi Wang[2], Hongzhi Li[3], Sheela Pangeni Pokharel[1], Sandra Li[1], Mingli Li[1], Xiaobao Xu[1], Wei Lu[1], Qiao Liu[1], Nicole Mattson[1], Kevin Yining Chen[2], Jinhui Wang[3], Yate-Ching Yuan[3], David Horne[3], Steven T. Rosen[3], Yadira Soto-Feliciano[4], Zhaohui Feng[2], Takayuki Hoshii [2], Gang Xiao [1,5], Markus Müschen [1,6], Jianjun Chen [1,3], Scott A. Armstrong [2,8✉] & Chun-Wei Chen [1,2,3,8✉]

Identification of novel functional domains and characterization of detailed regulatory mechanisms in cancer-driving genes is critical for advanced cancer therapy. To date, CRISPR gene editing has primarily been applied to defining the role of individual genes. Recently, high-density mutagenesis via CRISPR tiling of gene-coding exons has been demonstrated to identify functional regions in genes. Furthermore, breakthroughs in combining CRISPR library screens with single-cell droplet RNA sequencing (sc-RNAseq) platforms have revealed the capacity to monitor gene expression changes upon genetic perturbations at single-cell resolution. Here, we present "sc-Tiling," which integrates a CRISPR gene-tiling screen with single-cell transcriptomic and protein structural analyses. Distinct from other reported single-cell CRISPR screens focused on observing gene function and gene-to-gene/enhancer-to-gene regulation, sc-Tiling enables the capacity to identify regulatory mechanisms within a gene-coding region that dictate gene activity and therapeutic response.

[1] Department of Systems Biology, Beckman Research Institute, City of Hope, Duarte, CA, USA. [2] Department of Pediatric Oncology, Dana-Farber Cancer Institute—Harvard Medical School, Boston, MA, USA. [3] City of Hope Comprehensive Cancer Center, Duarte, CA, USA. [4] Rockefeller University, New York, NY, USA. [5] Department of Immunology, Zhejiang University School of Medicine, Hangzhou, China. [6] Yale School of Medicine, New Haven, CT, USA. [7] These authors contributed equally: Lu Yang, Anthony K. N. Chan, Kazuya Miyashita. [8] These authors jointly supervised this work: Scott A. Armstrong, Chun-Wei Chen. ✉email: scott_armstrong@dfci.harvard.edu; cweichen@coh.org

The integration of CRISPR (clustered, regularly interspaced, short palindromic repeats) with next-generation sequencing technology for high-throughput genetic screens is a powerful tool for discovering functional genes in various pathways and cellular contexts[1,2]. Furthermore, high-density CRISPR targeting of coding exons has been demonstrated to identify functional domains in genes[3–7]. However, the traditional CRISPR dropout/enrichment screens restricted the application to investigate functional elements associated with cell survival phenotypes. Recent breakthroughs in combining the CRISPR library screens with droplet RNA-sequencing (RNA-seq) platforms demonstrated the capacity of monitoring the gene expression changes upon genetic perturbations in single cells (e.g., Perturbseq, CRISP-seq, CROP-seq)[8–11]. The current single-cell CRISPR screens focused on observing single-gene function, gene-to-gene interaction, and enhancer-to-gene regulation[10,12–16]. Nevertheless, the potential of single-cell CRISPR screen technology to examine the gene function at a sub-gene resolution has not been fully explored.

In this study, we develop a single-cell CRISPR gene tiling pipeline "sc-Tiling" to provide high-resolution transcriptomic profiling of the coding regions of histone H3 lysine 79 (H3K79) methyltransferase DOT1L, an epigenetic therapeutic candidate selectively essential to mixed-lineage leukemia gene-rearranged (MLL-r) leukemia[17–19]. Furthermore, we couple the sc-Tiling with three-dimensional structural modeling and discovered a previously unrecognized self-regulatory domain in DOT1L that modulates the chromatin interaction, enzymatic activation, and therapeutic sensitivity in MLL-r leukemia.

## Results

### Development of the sc-Tiling screen.
Recent achievements in cancer epigenetics include discovery of a central role for the H3K79 methyltransferase DOT1L in maintaining MLL-r leukemia, an aggressive malignancy recognized in 5–10% of human acute leukemia cases[19,20]. A selective DOT1L inhibitor, EPZ5676 (Pinometostat)[21], has demonstrated proof-of-principle clinical benefits via induction of differentiation of MLL-r leukemic cells in a phase I clinical trial[22]. However, the variable responses of patients with MLL-r in this trial underscore the need for additional mechanistic insights into functional regions of DOT1L to improve therapeutic efficacy and trial designs for DOT1L-targeted therapy.

To achieve high-resolution characterization of DOT1L's function, we developed a single-cell CRISPR gene-tiling approach named sc-Tiling, which utilizes a capture sequence (CS1: 5′-GCTTTAAGGCCGGTCCTAGCA-3′) at the end of each single guide RNA (sgRNA) for direct capture by the Chromium Next GEM Single Cell 3′ Kit v3.1 (Fig. 1a and Supplementary Fig. 1a–c)[11]. We cloned a pool of 602 sgRNAs that target most of the "NGG" protospacer adjacent motifs within the mouse Dot1l coding exons (average targeting density 7.7 bp per sgRNA; Supplementary Fig. 2a, b and Supplementary Data 1). We then delivered this CRISPR library into Cas9-expressing mouse MLL-AF9 transduced leukemic cells (MLL-AF9-Cas9+; Supplementary Fig. 2c), a well-established murine leukemia model that mimics human MLL-r conditions[18,23]. Three days after transduction (Supplementary Fig. 2d, e), the cells carrying library constructs were subjected to droplet single-cell barcoding and messenger RNA (mRNA)/sgRNA library preparation using the 10X Chromium workflow (Fig. 1a). Subsequent single-cell transcriptomic analysis revealed an average of 26,350 reads per cell and a median of 2935 genes detected per cell (Supplementary Fig. 3). To avoid contamination by doublets and multi-sgRNA-infected cells, we filtered out any single cells carrying more than one sgRNA

sequence. Finally, 88.2% of single cells (4362 out of 4943) passed the quality control (QC) filter (Fig. 1b), giving an average library coverage of 7.1 cells per sgRNA.

Single-cell projections using Uniform Manifold Approximation and Projection (UMAP)[24] of DOT1L-dependent genes[18] identified seven cell clusters (Fig. 1c). Gene expression annotation revealed distinct distributions of cells expressing leukemia-associated genes (Meis1, Hoxa9, and Myc; clustered toward the right) vs. myeloid-differentiation markers (Cd11b, Gr1, and Ltf; clustered toward the left) (Fig. 1d). Cells expressing sgRNAs targeting the functionally essential lysine methyltransferase (KMT) core (residues M127–P332; total 56 sgRNA) of DOT1L[17,25] clustered to regions that overlap with the differentiated myeloid population (Fig. 1e). On the contrary, the sgRNAs targeting a non-essential region of DOT1L (the C-terminal end 100 amino acids of DOT1L; total 54 sgRNA) behaved similarly to spiked-in negative control sgRNAs (targeting Firefly luciferase [Luc], Renilla luciferase [Ren], green fluorescent protein [GFP], red fluorescent protein [RFP], and Rosa26 coding sequences; Supplementary Data 1), with both clusters to the region representing undifferentiated leukemia (Supplementary Fig. 4a). Trajectory analysis (pseudo-time)[26] correlated closely with the expression of these marker genes, with leukemia-associated genes being gradually reduced, while myeloid-differentiation markers increased along the pseudo-time trajectory (Fig. 1f, right to left and Supplementary Fig. 4b). These results indicate efficient CRISPR editing of DOT1L in cells expressing the CS1 direct-capturable sgRNA library.

### Structural and transcriptomic profiling of sc-Tiling.
To evaluate the resolution of sc-Tiling for detecting functional elements within a protein domain, we summarized the overall behavior of neighboring sgRNAs using a local-smoothing strategy[5] (Fig. 1g), and mapped the smoothened pseudo-time score to a cryo-electron microscopy structure of the DOT1L KMT core in an "active state" interacting with a histone H2B-ubiquitinylated nucleosome (Fig. 1h)[27,28]. Our results revealed that within the KMT core domain, the resolution of sc-Tiling allowed recognition of all the amino acid residues that directly contacted the enzymatic substrate S-adenosyl methionine (SAM pocket) and the D1 loop (residues P133–T139[25]) (Supplementary Fig. 5a). This method also detected the critical regions within the KMT core domain that mediate its chromatin interaction. These include the W22–D32 loop (Supplementary Fig. 5b; interacts with histone H4 tail), R282 loop (Supplementary Fig. 5c; interacts with the histone H2A/H2B acidic patch), and T320–K330 helix (Supplementary Fig. 5d; interacts with the ubiquitin conjugated to histone H2BK120)[27,28]. Taken together, sc-Tiling clearly distinguished the functional regions of KMT from the non-essential region (residues A33–T100) that is not involved in substrate/ligand interaction, revealing the capacity of single-cell CRISPR gene-tiling to pinpoint functional elements at a sub-domain resolution.

To identify novel functional elements that modulate DOT1L activity, we utilized the top 100 genes affected by DOT1L inhibitor[18] to develop a high-resolution transcriptomic correlation heatmap across DOT1L protein (Fig. 2a). This method revealed two functionally distinct segments of DOT1L, i.e., the N-module (residues M1–T900) and the C-module (residues P901–N1537). The strong correlation of the sgRNAs targeting the C-module with the negative control sgRNAs (Supplementary Fig. 4a) indicates a lack of essential components in the C-terminal portion of DOT1L. On the other hand, we observed several functional regions of DOT1L within the N-module, including the KMT core (black dashed triangle)[25,27,28] and the AF9-binding motif (green dashed box; residues T863–T900)[29]. Whereas the

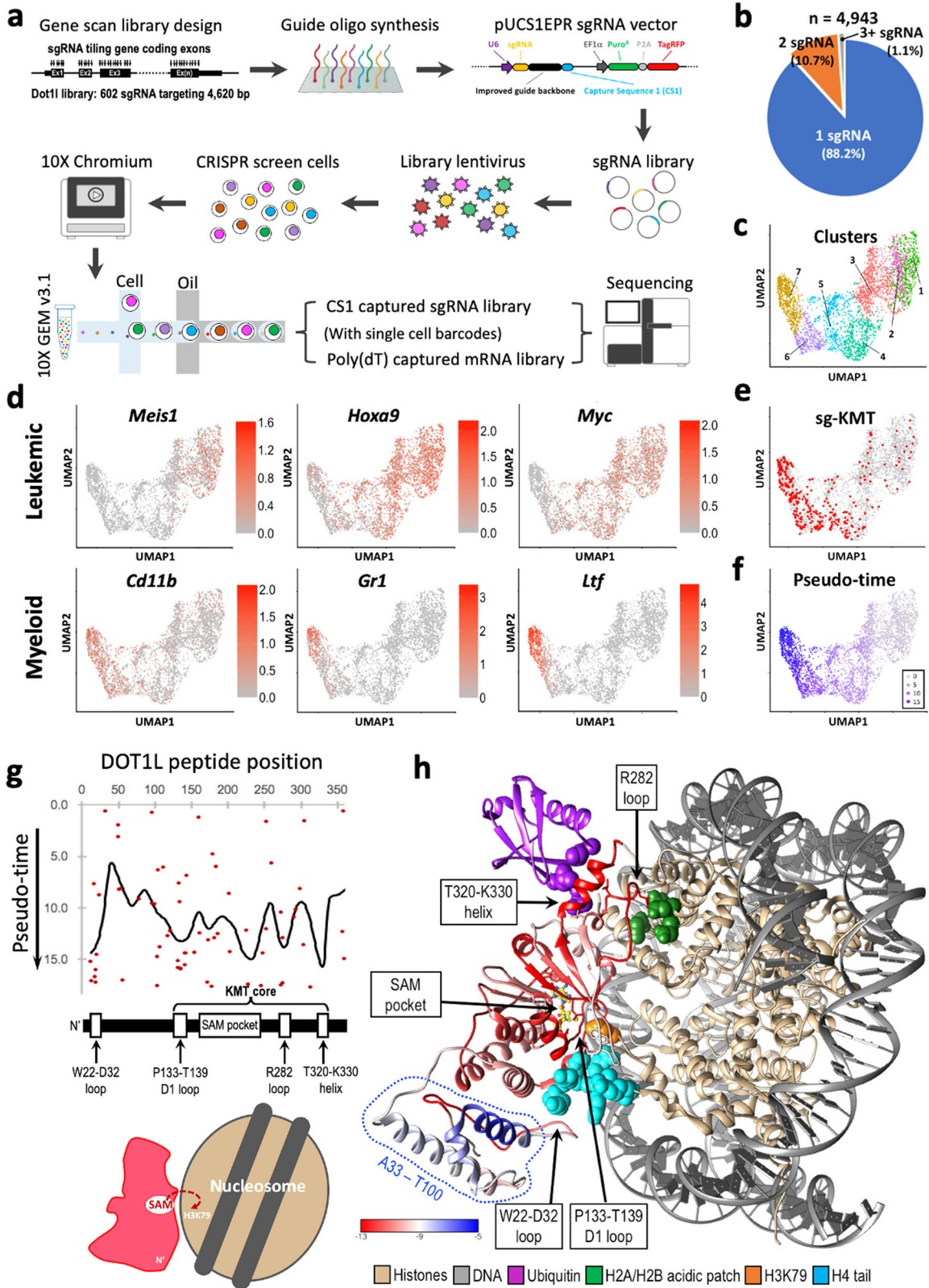

AF9-binding motif showed a moderate correlation (Pearson score ~0.75) with the KMT core, we identified a region (cyan dashed box; designated as the "R domain") located in the center of the N-module that exhibited a higher correlation (Pearson score >0.8) with the KMT core in the transcriptional signature. Based on this observation, we presumed that disrupting the function of the R domain would impair the survival of MLL-AF9 leukemia cells,

similar to inhibition of the KMT core. To test this, we utilized the DOT1L-tiling CRISPR library to perform pooled survival screens[4,6] in MLL-AF9-Cas9$^+$ cells and examined the cell survival by comparing the frequencies of each integrated sgRNA sequence before vs. after 3-, 6-, 9-, or 12-day cultures using high-throughput sequencing (Fig. 2b). Our results revealed a progressive depletion of clusters of sgRNAs (and smoothed

**Fig. 1 Single-cell CRISPR gene tiling of DOT1L. a** Schematic outline of sc-Tiling library construction and screening in MLL-AF9-Cas9[+] cells. **b** Assignment rates for direct-capture sgRNA. The total number of cells and fraction of cells assigned a single guide, two guides, or more than two guides are indicated. **c** Two-dimensional projection (UMAP) of cell clusters based on sc-RNAseq of DOT1L-dependent genes. The transcriptionally distinguishable cell populations (1–7) are color labeled. **d** Annotation of leukemia-associated (*Meis1*, *Hoxa9*, and *Myc*) and myeloid-differentiation (*Cd11b*, *Gr1*, and *Ltf*) gene expression on UMAP. **e**, **f** Annotation of **e** cells harboring sgRNAs targeting the DOT1L KMT core (red) and **f** pseudo-time value (purple gradient) on UMAP. **g** Median pseudo-time of each sgRNA constructs (dots) and the smoothed pseudo-time score (line) of the KMT core. **h** Three-dimensional annotation of smoothed pseudo-time score relative to a cryo-EM structural model of "active state" DOT1L (residues M1–P332) bound to a ubiquitylated nucleosome (PDB ID: 6NQA; and a simplified scheme shown on the bottom-left)[27]. Histones (gold; including H2A, H2B, H3, and H4), DNA (gray), ubiquitin (purple; conjugated to histone H2BK120; DOT1L contact points on ubiquitin are labeled as purple spheres), histone H4 N-terminal tail (cyan spheres), the enzymatic substrate SAM (colored sticks), histone H3K79 (orange spheres), and an H2A/H2B acidic patch (green spheres) are shown. Enlarged images are shown in Supplementary Fig. 5. Source data are available in the Source Data file.

---

CRISPR scan scores) targeting the KMT core, AF9-binding motif, and the first half of the R domain (designated as the "R1 element;" residues F460–G555). Furthermore, we sought to combine the principal component analysis of sc-Tiling (PC1 score) with the survival CRISPR scan score for individual amino acids in DOT1L (Fig. 2c). This approach revealed that the distribution of KMT core (black dots) overlaps with a segment located in the center of the R1 element (R1 center; E489–L515; red dots) in both transcriptomic and survival profiling, suggesting the functional association of this region with the KMT core.

**R domain modulates the efficacy of DOT1L inhibitory therapy**. To confirm the results from sc-Tiling analyses, we chose three sgRNAs each targeting the KMT core, AF9-binding motif, and the R1 center for functional validation (guide sequence and editing efficiency shown in Supplementary Fig. 6). Using an RFP flow cytometric growth competition assay (Supplementary Fig. 7)[3] and immunoblotting, we observed that compared to the sgRNAs targeting AF9-binding motif, expression of sgRNAs targeting the R1 center resulted in a more drastic suppression of cell proliferation (Fig. 2d) and impaired histone H3K79 methyltransferase activity (Fig. 2e), resembling the effects of sgRNAs targeting the KMT core. In addition to a similar UMAP distribution between cells expressing sgRNAs targeting the KMT core and the R1 element (Fig. 2f), single-cell droplet RNA seq (sc-RNA-seq) revealed significantly overlapped gene regulation between these two sgRNA-targeted populations (Fig. 2g, h). These results indicate functional coordination between the DOT1L KMT core and R1 element for histone modification.

To investigate whether the R domain mediates the response of MLL-AF9 leukemia cells to DOT1L-inhibitory treatment, we compared a pair of pooled survival tiling screens conducted under control (dimethyl sulfoxide (DMSO)) vs. DOT1L-inhibited (1 µM EPZ5676) conditions (Fig. 3a). Consistent with the results of the sc-Tiling, we observed that a cluster of 27 sgRNAs targeting the R1 region (residues F460–G555) sensitized the MLL-AF9-Cas9[+] cells to DOT1L inhibition (Fig. 3b and Supplementary Fig. 8c). By contrast, a cluster of 36 sgRNAs targeting the residues A558–C662 (designated as the "R2 element") exhibited a significantly increased CRISPR score only in the DOT1L-inhibited condition (Fig. 3a, b). The expression of individual sgRNAs targeting the R2 element exhibited minimal impact on the proliferation of MLL-AF9-Cas9[+] cells (Fig. 3c and Supplementary Fig. 8a, b), but increased the resistance index to the DOT1L inhibitor (Fig. 3d and Supplementary Fig. 8c), confirming the EPZ5676-resistant phenotype we observed in the CRISPR gene body scans. Computational modeling of the R domain (residues F460–C662) revealed a consensus "coiled-coil" structure consisting of four alpha-helices (Fig. 3e), which is capable of interacting with the KMT core domain of DOT1L (Fig. 3f). Within the R domain, the R1 element (consisting of CC0 and CC1) overlaps with an area previously reported to interact with

AF10[30,31], a coactivator of DOT1L required for methyltransferase activation. On the other hand, the R2 element (consisting of CC2 and CC3) is predicted to interact with the DOT1L KMT core and masks the R282 loop (Fig. 3f), thereby interrupting the DOT1L–nucleosome interaction and methyltransferase activity of the KMT core. This model suggests that the R domain mediates the transition from a "closed" to an "open" state of DOT1L (Fig. 3g; left to right), which is required before the engagement of the KMT core with nucleosomes for H3K79 methylation (Fig. 3g; blue area summarized in Fig. 1h).

To evaluate the impact of this self-regulatory mechanism on DOT1L-targeted therapy, we queried the cBioPortal database[32] and focused on the R2 element (residues A558–C662) that exerted a robust EPZ5676-resistant phenotype in the CRISPR scan. Out of a total of 54,510 patient samples, we found 19 DOT1L variant alleles to exist in this 105-amino acid region (Supplementary Fig. 9a and Supplementary Table 1). Compared to the expression of wild-type-DOT1L constructs, the expression of several mutant-DOT1L constructs (each harbors a single amino acid missense mutation) in MLL-AF9 cells resulted in an increased resistance to EPZ5676 treatment (Fig. 3h and Supplementary Fig. 9b, c). We then focused on the top three drug-resistant variants (Q584P, L626P, and C637G) and found that these mutant-DOT1L led to an elevated H3K79me2 (Supplementary Fig. 10) and required a higher dosage of EPZ5676 to suppress their activity compared to wild-type-DOT1L (Fig. 3i, j). Computational modeling of these drug-resistant variants indicates that mutations at these residues may destabilize alpha-helix bundles and lead to dissociation of the R domain from the KMT core, resulting in increased kinetic activity and tolerance to DOT1L-inhibitory therapy (Supplementary Fig. 11).

## Discussion

High-throughput CRISPR genetic screens have been wildly used for discovering functional genes in mammalian systems. In contrast, the potential of CRISPR technology to investigate gene function at a sub-gene (i.e., protein domain or sub-domain) resolution has not been fully explored. Furthermore, traditional pooled CRISPR screens limit the ability to identify functional elements associated with cell killing/proliferation phenotypes (i.e., by observing the depletion or enrichment of specific sgRNA). The requirement for significant changes in cell number in survival CRISPR screens (which typically take 2–4 weeks of culture) prohibits the determination of causal mechanisms induced by CRISPR perturbation.

To overcome this obstacle, our study integrated a CRISPR gene-tiling screen with a recently available direct-capture Perturb-seq workflow[11] to develop the single-cell CRISPR gene body-scan pipeline sc-Tiling. Using this approach, we provide a high-resolution transcriptomic correlation map across DOT1L, an epigenetic therapeutic candidate essential to *MLL*-r

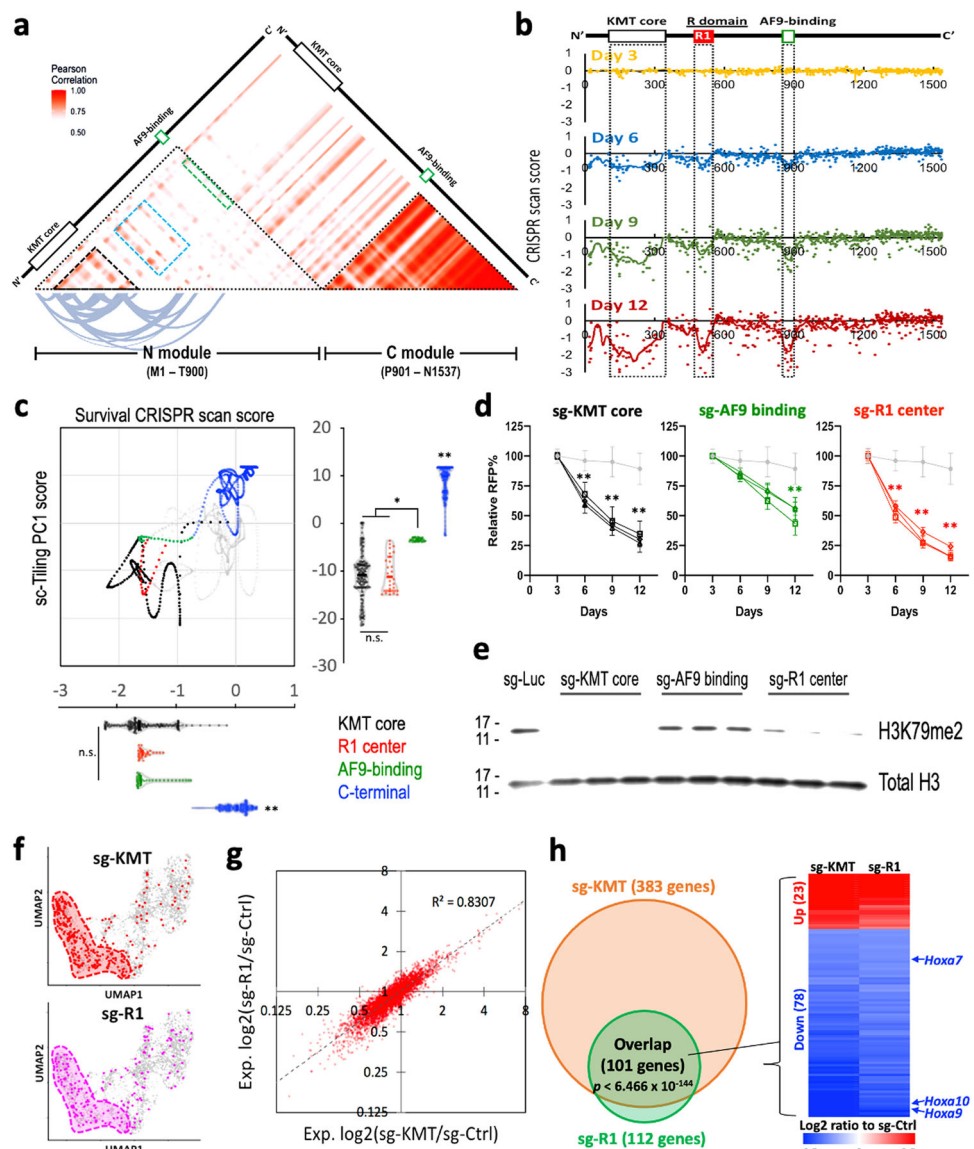

**Fig. 2 sc-Tiling pinpoints functional elements in DOT1L. a** Heatmap depicts Pearson correlations between sgRNAs targeting different positions across the DOTL1 protein. The curved lines indicate highly correlative (Pearson score >0.8) residue pairs in the N-module of DOT1L. **b** CRISPR scan score of each sgRNA (dots) and smoothed score (line) of the DOT1L-tiling survival screen in MLL-AF9-Cas9[+] leukemia at the indicated number of days in culture. **c** Combinational analysis of the sc-Tiling principal component 1 (PC1; *y*-axis) and survival CRISPR scan score (day 12; *x*-axis) of individual amino acid residues (dots) in DOT1L. Residues compose of R1 center (E489–L515; red dots) correlate with the KMT core (black dots) at both the transcriptomic and cellular survival phenotypes. *$P < 0.01$ by two-sided Student's *t* test. **$P < 0.01$ to all other groups. **d** Effect of individual sgRNAs targeting the KMT core (black; three independent sgRNAs), AF9-binding motif (green; three independent sgRNAs), or R1 center (red; three independent sgRNAs) of DOT1L on the proliferation of MLL-AF9-Cas9[+] leukemia. Data represent mean ± 95% confidence interval of a quadruplicate experiment. **$P < 0.001$ by two-sided Student's *t* test compared to a sgRNA targeting Luciferase (sg-Luc; gray). **e** Western blot of H3K79me2 and total histone H3 in MLL-AF9-Cas9[+] cells expressing indicated sgRNAs (three independent sgRNAs per domain). **f** Annotation of cells harboring sgRNAs targeting the KMT core (red) or R1 element (pink) on UMAP. **g** Correlation of gene expression changes induced by sgRNAs targeting the KMT core (*x*-axis) and R1 element (*y*-axis) summarized from the sc-Tiling of DOT1L. **h** Overlap of differentially expressed genes in cells harboring sgRNAs targeting the KMT core (orange) and R1 element (green), including the known DOT1L-driven leukemia genes *Hoxa7*, *Hoxa9*, and *Hoxa10*. n.s. Not significant. Source data are available in the Source Data file.

leukemia[17–19]. We noted that the traditional survival CRISPR gene scan (*x*-axis; Fig. 2c) was unable to distinguish the AF9-binding motif (mediates recruitment of AF9-containing super elongation complex to support gene transcription)[33,34] from the KMT core (mediates H3K79 methylation and open chromatin)[18]. In contrast, sc-Tiling (*y*-axis; Fig. 2c) efficiently differentiated these two functionally distinct domains through transcriptional profiling. The fact that cell killing through targeting the AF9-binding motif does not impair the H3K79me2 level (Fig. 2e) testifies the catalytic-independent role of the AF9-binding motif

in DOT1L. We envision a significant advance of sc-Tiling to recognize underlying mechanisms of the functional domains. Furthermore, we foresee the transcriptomic profiling in sc-Tiling to enable dissection of functional elements that participate in diverse cellular processes (e.g., metabolism, cell fate decision, tissue homeostasis) that the end phenotypes might not be the cellular survival or proliferation.

Although the limitations of CRISPR genome editing (e.g., variable cutting efficiency, potential for off-targeting, and the mosaic effect [i.e., generation of random mutations]) remain

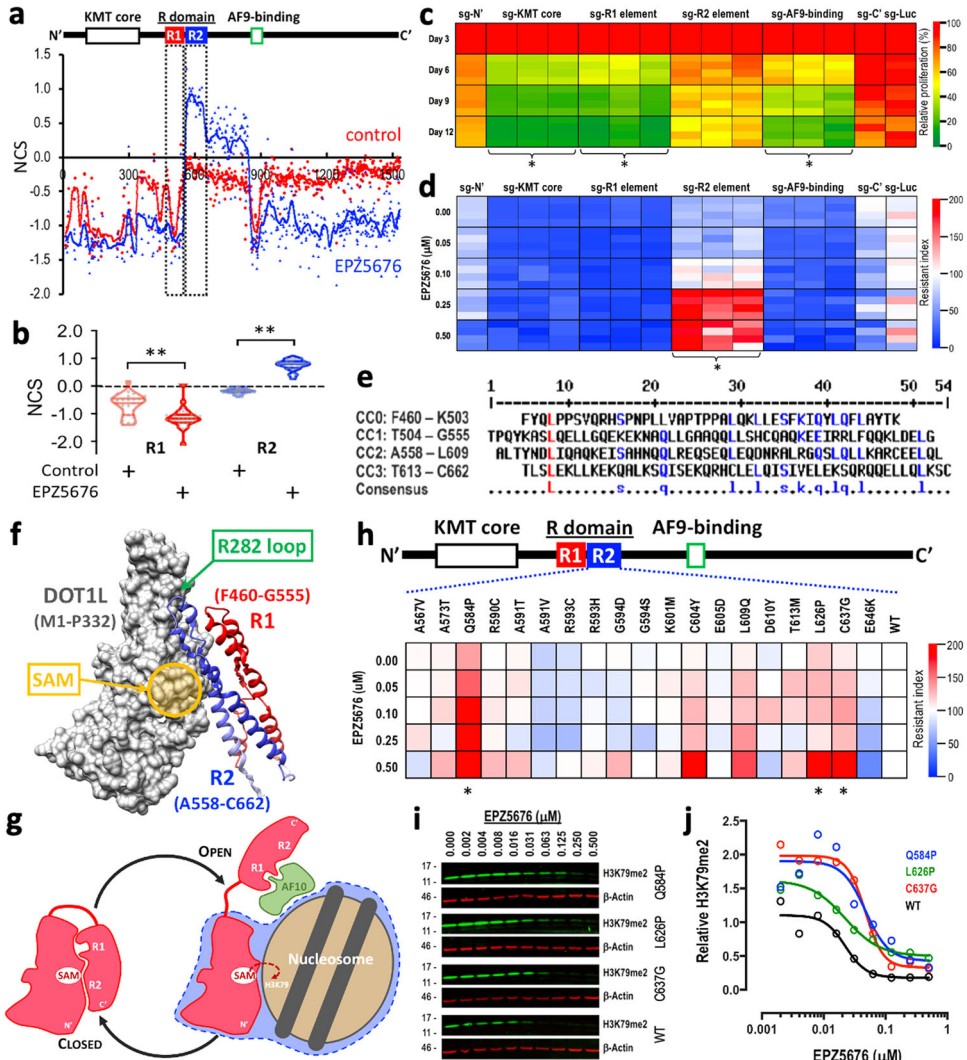

**Fig. 3 sc-Tiling identifies noncanonical EPZ5676-resistant alleles in the human population. a** Normalized CRISPR score (NCS) of each sgRNA construct (dots) and the smoothed score (line) of the pooled DOT1L-tiling survival screen before vs. after 12 days of treatment in control (red) or 1 μM EPZ5676-treated (blue) MLL-AF9-Cas9+ leukemia cells. Data represent the average of a triplicate experiment. **b** Violin dot plots showing the NCS of each sgRNA targeting the R1 (red; 27 sgRNAs) and R2 (blue; 36 sgRNAs) elements in control (DMSO) or 1 μM EPZ5676-treated MLL-AF9-Cas9+ leukemia cells. **P < 0.001 by two-sided Student's t test. **c** Heatmap showing the effect of individual sgRNAs targeting the indicated areas of DOT1L (Supplementary Fig. 6) on the proliferation of MLL-AF9-Cas9+ leukemia cells on days 3, 6, 9, and 12. Data represent the observed values of a quadruplicate experiment. *Significantly (P < 0.01 by two-sided Student's t test) more depletion compared to the sgRNA targeting a non-essential N′ region (sg-N′) on day 12. **d** Heatmap showing EPZ5676 resistance index of MLL-AF9-Cas9+ leukemia cells transduced with sgRNAs targeting the indicated areas of DOT1L (Supplementary Fig. 6). Data represent the observed values of a quadruplicate experiment. *Significantly (P < 0.01 by two-sided Student's t test) higher resistance compared to sg-N′ at 0.5 μM EPZ5676. **e** Peptide sequence alignment of the R domain (residues F460–C662) in human DOT1L. The predicted alpha-helices in this coiled-coil domain are designated CC0–CC3 and the consensus residues between the helixes are noted. **f** Computationally modeled structure of the human DOT1L R1 (red) and R2 (blue) coiled-coil domains interacting with the KMT core domain (gray; PDB ID: 3UWP)[46]. **g** Cartoon representation of the R1/R2 self-regulatory module mediating the closed (left) vs. open (right) states of DOT1L. **h** Heatmap showing EPZ5676 resistance index of MLL-AF9 leukemia cells transduced with human DOT1L cDNA harboring clinically observed variants (from cBioPortal database) in the R2 element. Data represent the averaged values of a quadruplicate experiment. *Significantly (P < 0.01 by two-sided Student's t test) higher resistance compared to wild-type at 0.5 μM EPZ5676. **i** Western blot images of H3K79me2 (green) and β-actin (red) and (**j**) quantitative measurement of relative H3K79me2 level (normalized to β-actin) in MLL-AF9 leukemia cells transduced with wild-type (WT; black), Q584P (blue), L626P (green), or C637G (red) human DOT1L cDNA. Cells were treated with EPZ5676 for 3 days. Source data are available in the Source Data file.

concerns in the CRISPR sc-Tiling approach, by considering multiple sgRNAs clustered in a peptide region via a local-smoothing strategy, we significantly increased the statistical confidence and minimized the impact of noise associated with individual sgRNAs. Importantly, the use of single-cell transcriptional profiling in sc-Tiling could predict functional elements and corresponding gene regulations that led to a cellular survival phenotype after prolonged culture, and provided superior

resolution in detecting sub-domain functional elements than survival CRISPR gene-tiling screens using pooled sequencing (i.e., Fig. 1g vs. 2b; KMT core). Furthermore, when we coupled sc-Tiling with three-dimensional structural modeling, we discovered a self-regulatory R domain in DOT1L that modulates chromatin interaction, enzymatic activation, and therapeutic sensitivity in *MLL*-r leukemia. To our knowledge, this is the first characterization of an intragenic regulatory module that mediates

switching between a "closed" and an "open" state of an epigenetic enzyme.

Finally, our study demonstrates the utility of combining sc-Tiling with consortium genomic databases (e.g., cBioPortal, CCLE, dbSNP; Supplementary Table 1 and Supplementary Fig. 12) for de novo identification of therapeutically relevant alleles in the human population (Fig. 3h). We propose that sc-Tiling may complement the rapidly growing multi-omics databases to provide additional insights that bridge functional genomics, structural biology, and clinical investigation. We envision that this approach will accelerate the recognition of clinically impactful variants within the human genome and has the potential to direct more precise clinical trials and therapeutic decisions.

## Methods

**Cas9-expressing MLL-AF9 leukemic cell culture.** Mouse MLL-AF9 leukemic cells were generated by transformation of mouse bone marrow Lin−Sca1+cKit+ cells with a MIG (MSCV-IRES-GFP) retrovirus expressing the MLL-AF9 fusion protein and transplanted into sublethally irradiated recipient mice[23]. Leukemic blasts were subsequently harvested from the diseased mice and cultured in vitro in Iscove's modified Dulbecco's medium (Gibco) plus 15% fetal bovine serum (Gibco) supplemented with 20 ng/ml mouse stem cell factor (PeproTech), 10 ng/ml mouse interleukin-3 (IL-3) (PeproTech), 10 ng/ml mouse IL-6 (PeproTech), penicillin (100 U/ml; Gibco), streptomycin (100 μg/ml; Gibco), and plasmocin (5 μg/ml; InvivoGen). Cas9-expressing MLL-AF9 cells were established through lentiviral transduction of LentiCas9-Blast (Addgene)[35], followed by blasticidin S (10 mg/ml; Gibco) selection, single-cell cloning, and CRISPR editing efficiency test (Supplementary Fig. 2c, d).

**CRISPR gene-tiling screens.** sgRNA sequences targeting the coding regions of mouse Dot1l (Supplementary Data 1) were designed using the Genetic Perturbation Platform (Broad Institute)[36]. Briefly, sgRNA oligonucleotides were synthesized via microarray (CustomArray) and cloned into the pUCS1EPR lentiviral sgRNA vector (Supplementary Fig. 1a) using BsmBI (NEB)[36]. A step-by-step protocol describing the cell culture protocol can be found at Protocol Exchange[37]. The sgRNA library was packaged by HEK293 cells (ATCC) cotransfected with psPAX2 (Addgene) and pMD2.G (Addgene) to produce lentiviral particles, and pre-titrated to obtain 10–20% infection (monitored by flow cytometry for RFP [tagRFP] expression) in the MLL-AF9-Cas9+ cells. Each screen culture was maintained in at least 1000× the number of constructs in each library. For sc-Tiling, library-transduced cultures were selected using puromycin (2.5 μg/ml; Gibco) for 3 days and subjected to single-cell separation and barcoding using a Chromium Controller (10X Genomics). For survival CRISPR gene tiling, the sgRNA library-transduced cells were subcultured every 3 days for a total of 12 days. At each designated time point, the number of cells from cultures that covered at least 1000× the number of constructs in the library was collected for analysis.

**sc-Tiling data analysis.** Using the Next GEM Single Cell 3′ Kit v3.1 and a Chromium Controller (10X Genomics), CS1-captured sgRNA and the poly(dT)-captured mRNA from each single cell were converted to next-generation sequencing libraries (Supplementary Fig. 1c), and sequenced (paired-end 150 base pair) using Illumina HiSeqX (Novogene Inc.). Sequencing QC and data preprocessing were performed using Seurat v3.0[24]. Low-quality single cells with abnormal gene numbers (<200 or >4500) or significant mitochondrial RNA contamination (>10% reads) were removed (Supplementary Fig. 3a). The normalized expression data from selected single cells then underwent dimensionality reduction by principal component analysis and UMAP embeddings for visualization and clustering. Cells were clustered based on the poly(dT)-captured transcriptome information and simultaneously annotated by CS1-captured sgRNA. Single cells with more than one detected sgRNA sequence (due to multiple sgRNA transductions or multiple cells in a single-cell droplet) were excluded. Pseudo-time trajectory analysis of the DOT1L inhibitor-affected genes was performed on single-cell transcriptomic data using Monocle[26]. Position-ordered Pearson correlation matrix across the Dot1l gene body was calculated based on the top 100 genes affected by DOT1L inhibition.

**Three-dimensional protein structural annotation of sc-Tiling.** First, the median value pseudo-time projection generated from sc-Tiling was summarized for each sgRNA. To depict the pseudo-time score over regions with no sgRNA coverage, we interpolated the signal via Gaussian kernel smoothing in R[38]. The bandwidth was defined by the maximum gap length of the non-covered regions for local smoothing due to regional uneven sgRNA densities. To map the smoothed pseudo-time score to peptide positions, the average pseudo-time score over the trinucleotide codons was calculated for each peptide position. Pairwise alignments of primary amino acid sequences were performed using CLC Main Workbench version 8.1 (Qiagen) to ensure functional annotations of the smoothed pseudo-time

scores of mouse Dot1l sc-Tiling data onto human DOT1L protein structures. Atomic data of macromolecular structures were retrieved from the Research Collaboratory for Structural Bioinformatics Protein Data Bank (RCSB Protein Data Bank (PDB) at https://www.rcsb.org)[39] in PDB file format. The PDB files were visualized and analyzed using UCSF Chimera (version 1.14 build 42000)[40]. Subsequently, the smoothed pseudo-time scores were mapped onto three-dimensional protein structures using the "Defined Attribute" and "Render by Attribute" functionalities in UCSF Chimera[40].

**Survival CRISPR gene tiling data analysis.** Genomic DNA from survival screen cell pellets was harvested, PCR-amplified (NEBNext Ultra II Q5; NEB) using primers DCF01 5′-CTTGTGGAAAGGACGAAACACCG-3′ and CS1_R01 5′-TGCTAGGACCGGCCTTAAAGC-3′ (Supplementary Fig. 1a and Supplementary Table 2), and subjected to high-throughput sequencing (NextSeq550, Illumina). To quantify sgRNA reads in the library, we first extracted 20-nucleotide sequences that matched the sgRNA backbone structure (5′-CACCG and 3′-GTTT) from raw fastq reads. Extracted reads were then mapped to a reference database built from corresponding sgRNA library sequences using Bowtie2[41]. Only reads that perfectly matched the reference database were counted. The frequency for individual sgRNAs was calculated as the read counts of each sgRNA divided by the total read counts matched to the library. Individual sgRNAs with read counts <5% of the expected frequency were excluded from downstream analysis. A CRISPR score was defined as a log 10-fold change in the frequency of individual sgRNAs between early (day 0) and late (defined time points) of the screened samples, calculated using the edgeR R package[42] based on the negative binomial distribution of sgRNA read count data. To obtain a CRISPR scan score over regions with no sgRNA coverage, we interpolated the signal via Gaussian kernel smoothing in R[38]. Bandwidth was defined as the maximum gap length of the non-covered regions for local smoothing due to regional uneven sgRNA densities. To map CRISPR scan scores to peptide positions, the average CRISPR scan score over the trinucleotide codons was calculated for each peptide position. To compare survival screens performed in different culture conditions (e.g., control vs. EPZ5676-treated), the smoothed CRISPR scan score was further normalized by the median CRISPR score of the negative control sgRNA (defined as 0.00; sgRNA targeting Luc, Ren, GFP, RFP, and Rosa26) and the median CRISPR score of the positive control sgRNA (defined as −1.00; sgRNA targeting mRpa3)[3] within the screen data.

**Computational structural modeling.** Four helices (CC0–CC3) of the R domain were predicted using the PSIPRED v3.3 server[43]. Sequence alignment of the helical regions (Fig. 3e) was produced using the MultAlin v5.4.1 server[44]. The model of the coiled-coil domain was predicted using the I-TASSER server[45]. The complex model of the R domain and KMT core domain (PDB ID: 3UWP)[46] was picked from 5000 complex models generated using the ZDOCK v3.0.2 software[47]. The best model (Fig. 3f) was selected based on the largest number of hydrophobic contact residue pairs between the KMT core and R domain. The structures were visualized using the PyMOL v1.8.6 software (Schrödinger, LLC) and UCSF Chimera[40].

**Generation of human DOT1L variant cDNA expression constructs.** A MIY (MSCV-IRES-YFP) retroviral construct expressing wild-type human DOT1L and yellow fluorescent protein (YFP) was obtained from Dr. Yi Zhang[25]. The initial wild-type human DOT1L complementary DNA (cDNA) (MIY-DOT1L-WT) was then point-mutated to obtain 19 clinically observed DOT1L variants (Supplementary Fig. 9b) using the Q5 Site-Directed Mutagenesis Kit (NEB). The mutated DOT1L cDNA fragments were confirmed using Sanger sequencing (Eton Bioscience).

**Western blotting.** Cells were harvested and lysed in LDS sample buffer (Invitrogen) at $5 \times 10^6$ cells/mL, separated electrophoretically using Bolt 4–12% Bis-Tris plus gels (Invitrogen), and transferred onto polyvinylidene difluoride (PVDF) membranes (0.2 μm pore size, low fluorescence) using PVDF Mini Stacks and iBlot 2 (Invitrogen). Membranes were probed with rabbit anti-H3K79me2 antibody (D15E8, Cell Signaling Technology; 1:1000), rabbit anti-histone H3 (ab1791, Abcam; 1:10,000), and mouse anti-β-actin antibody (ab8226, Abcam; 1:1000) at 4 °C overnight. After washing, the membranes were incubated with horseradish peroxidase-linked goat anti-rabbit IgG antibody (CST7074, Cell Signaling Technology; 1:10,000), donkey anti-rabbit IgG antibody conjugated with Alexa Fluor 488 (ab150061, Abcam; 1:10,000), or donkey anti-mouse IgG antibody conjugated with Cy3 (AP192C, Sigma-Aldrich; 1:10,000) at room temperature for 1 h. Chemiluminescent signals were developed using the SuperSignal West Femto Substrate (Cat# 34095, Thermo Fisher). The chemiluminescent and fluorescent signals on Western blot membranes were detected using a ChemiDoc imaging system (Bio-Rad). Signal intensity from image files was analyzed using the ImageJ software (National Institutes of Health). Representative Western blot images were selected from at least two independently performed experiments.

**Growth competition assay.** Cas9-expressing MLL-AF9 cells were virally transduced with the designated constructs (RFP+ ipUSEPR lentiviral sgRNA constructs listed in Supplementary Fig. 6; YFP+ MIY retroviral DOT1L variant cDNA constructs listed in Supplementary Fig. 9) in 96-well plates at ~50% infection and monitored using flow cytometry for RFP or YFP (FP). At each time point, live cell

counts and the percentage of FP$^+$ cells (FP%) were obtained by high-throughput flow cytometry and 4′,6-diamidino-2-phenylindole (Invitrogen) dye exclusion using an Attune NxT flow cytometer with an autosampler (Thermo Fisher).

The relative proliferation (RP) of FP$^+$ (sgRNA- or DOT1L cDNA-expressing) vs. FP$^-$ (non-transduced) cells was defined as:

$$\text{Relative proliferation(RP)} = \frac{[N(t) \times \text{FP\%}(t)] \times [N(\text{d3}) \times (100 - \text{FP\%}(\text{d3}))]}{[N(\text{d3}) \times \text{FP\%}(\text{d3})] \times [N(t) \times (100 - \text{FP\%}(t))]} \quad (1)$$

where $N(t)$ and FP%$(t)$ are the observed live cell number and FP$^+$% at time point $t$; d3 denotes the day 3 time point.

The resistance index was defined as:

$$\text{Resistance index} = \frac{\text{RP}(x, m)}{\text{RP}(\text{con}, m)} \times 100\% \quad (2)$$

where RP$(x,m)$ is the RP of cells expressing sgRNA or DOT1L cDNA variant $x$ under $m$ μM of EPZ5676 (Selleck Chemicals) on day 9; con denotes the sg-Luc or wild-type DOT1L cDNA.

**Reporting summary**. Further information on research design is available in the Nature Research Reporting Summary linked to this article.

## Data availability

The 10X Genomics single-cell CRISPR and RNA-seq data generated in this study have been deposited in the Gene Expression Omnibus database under accession code GSE174307. Three-dimensional protein structures (PDB ID 3UWP and 6NQA) were obtained from the Research Collaboratory for Structural Bioinformatics Protein Data Bank (RCSB PDB; https://www.rcsb.org)[39]. Consortium genomic information were obtained from cBioPortal (https://www.cbioportal.org)[32], Cancer Cell Line Encyclopedia (CCLE; https://portals.broadinstitute.org/ccle)[48], and dbSNP (https://www.ncbi.nlm.nih.gov/snp/)[49] databases. Additional data that support the findings of this study are provided in the Supplementary information and Source data files. Source data are provided with this paper.

## Code availability

The computational codes/tool packages used in this study are available at https://github.com/l0yang05/singleCell_CRISPR_10x (GitHub) and through other developers and venders, including Genetic Perturbation Platform (Broad Institute)[36], Seurat v3.2.3[24], Monocle2.14.0[26], Gaussian kernel smoothing in R[38], CLC Main Workbench version 8.1 (Qiagen), Bowtie2.3.5.1[41], edgeR package 3.28.1[42], PSIPRED v3.3 server[43], MultAlin v5.4.1 server[44], I-TASSER server v5.1[45], ZDOCK v3.0.2 software[47], PyMOL v1.8.6 software (Schrödinger, LLC), UCSF Chimera 1.15[40], ImageJ 1.8.0_172[50], PRALINE multiple sequence alignment[51], FlowJo v9, and Attune NxT v3.1.2 (Thermo Fisher).

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

## Acknowledgements

This work was supported by the American Society of Hematology (to C.-W.C.), Alex's Lemonade Stand Foundation (to C.-W.C.), Leukemia & Lymphoma Society (to J.C. and S.T.R.), and National Institutes of Health Grants CA176745, CA206963 (to S.A.A.), CA197498, CA233691, CA236626 (to C.-W.C.). The sequencing and structural computational studies were supported by the National Institutes of Health P30 award CA033572 (City of Hope). We thank Dr. Sarah Wilkinson for editing the manuscript.

## Author contributions

L.Y., A.K.N.C., K.M., C.D.D., X.W., S.P.P., M.L., X.X., Q.L., N.M., K.Y.C., J.W., Y.S.-F., Z. F., and G.X. performed the experiments; L.Y., A.K.N.C., H.L., S.L., W.L., Y.-C.Y., D.H., and C.-W.C. analyzed the data; D.H., S.T.R., T.H., M.M., J.C., S.A.A., and C.-W.C. provided conceptual input; L.Y., A.K.N.C., K.M., H.L., S.A.A., and C.-W.C. wrote the paper; S.A.A. and C.-W.C. conceived and supervised the study.

## Competing interests

The authors declare no competing interests.
