## [Peer Review File · Nature Communications]

Reviewers' Comments:

Reviewer #1:

Remarks to the Author:

General comment-summary:

Yang and colleagues combine CRISPR/Cas9-mediated editing with single cell RNA sequencing (that they call "sc-tiling") to dissect the role of the H3K79 methyltransferase DOT1L in leukemic cells. Hereby they established a library of >600sgRNAs (3' fused to a capture sequence) targeting the Dot1L ORF of 4.6kb, delivered by viral transduction into MLL-AF9 fusion gene immortalized mouse hematopoietic cells (Fig.1a-b). Sc-RNA sequencing revealed particular regions within the Dot1L ORF which resulted in differentiation or maintained an immature phenotype (Fig.1c-f). They found several regions in and outside the KMT domain that seemed essential to maintain the undifferentiated state (Fig.1g-h). They identified 3 critical regions in the N-terminal Dot1L ORF, including "R" between KMT and the AF9 binding domain (Fig.2). They also used the sc-tiling approach to identify regions that would modify cellular response to a well-characterized small molecule DOT1L-KMT inhibitor (EPZ5676). Hereby they observed that while the N-terminal part of R ("R1") enhanced, disrupting of the C-terminal part of R ("R2") impaired inhibitor activity. Mutational analysis revealed several critical residues in R2, and structural modeling suggested a regulatory mechanism of KMT activity involving R1 and R2 (Fig.3a-f). Interestingly, they identified multiple single-nucleotide variants in patient samples in the R2 region of DOT1L which, when expressed in MLL-AF9 expressing cells increased their resistance towards EPZ5676 (Fig.3h-j). Overall, this is a well-written and interesting paper showing how the combination of CRISPR/Cas9 editing with scRNA sequencing can help to better understand the structure/functional relations of a particular pharmaceutical target and the strategy of inhibition. Many findings that emerge from this technically sophisticated approach seem however of confirmatory nature to the current knowledge except the identification of a particular stretch of the DOT1L ORF that when mutated provides resistance against the EPZ5676. This observation raises interesting questions about the biological relevance of such variants/alterations for cancer/leukemia patients, particularly those that showed limited response to small molecule DOT1L inhibitors in clinical trials. A better characterization of the variants in this particular region would further increase the impact of this paper beyond the high technical achievements.

Specific comments:

1. The identification of a region "R2" that when altered leads to resistance to small molecule DOT1L inhibitors seems novel and interesting. It would be important to provide more information about these rare variants observed in 104/54510 "patient samples". What kind of samples were those? From cancer patients or other diseases? What kind of cancers? Were the variants identified upon de novo diagnosis or after chemotherapy? Can such variants also be acquired and/or selected during chemotherapy of human cancers? Are these variants present in human cancer cell lines? What about normal human beings: do we find DOT1L germline variants in this region?
2. Fig.1g needs to be better explained for non-experts. From far it appears to have more critical regions, particular in the KMT regions that reach a similar score, than the ones that are described in more detail in the structure in Fig.1h?
3. Fig.2a compare Dot1L ORF regions with impact on gene expression with the "AF9-binding motif" showing "a moderate correlation. However, in the experiments showed in Fig.3a, this region seemed very sensitive in control as well as EPZ5676 treated cells, in fact reaching the lowest NCS? The authors may clarify that in the text.

Reviewer #2:

Remarks to the Author:

In this manuscript, Lu et al integrated a CRISPR-tiling screen with the direct-capture Perturb-seq, named 'sc-tiling', to identify gene functions at a sub-gene resolution and investigate the regulatory mechanisms within gene-coding regions. Using this new approach, they discovered a novel

functional R domain in DOT1L that coordinated with KMT domain to regulate histone modification. It is interesting that the R1 and R2 elements in the R domain exhibit opposite responses during DOT1L-inhibitory treatment, indicating that this self-regulatory R domain dictates the response to DOT1L-targeted therapy.

Overall, this is a well-organized study with attractive novel points. This is the first report of an application of tiling screen at a single cell resolution. The discovery of the opposite response of the R1 and R2 is interesting too. Meanwhile, I have the following questions and suggestions that might be helpful for the refinement of the manuscript.

1. What are the pros and cons of sc-tiling compared to the survival-based pooled tiling screens? This is a critical question that needs to be addressed. A comparison of the sc-tiling result in Fig. 2A and the survival screen in Fig. 2B can be useful. I'd suggest performing a PCA analysis on the correlation matrix of Fig. 2A and to compare the principal component to the viability profile in Fig. 2B. This will tell us which amino acids are functionally associated with the KMT core at the transcriptome level, in comparison to those at the cell phenotypical level. More discussion on the pros and cons of the two alternative methods is preferred too.
2. In the sc-tiling screen, samples were collected at day 3 after transduction. Why this timepoint is chosen? Is this because DOT1L is very essential in MLL-AF9 cells? I wonder if day 3 is sufficient for some relatively inefficient sgRNAs to achieve genome editing. The authors should provide explanation and address the potential limitation.
3. As shown in Fig. 2D, the H3K79me2 level is decreased in sg-KMT and sg-R1 cells without DOT1L-inhibitory treatment. Does the H3K79me2 level alter upon sg-R2 or Q584P (Fig. 3I) without DOT1L-inhibitory treatment?
4. In Figures 2C and 2D, sg-Luc was used as the control. This doesn't introduce cleavage to the DNA. The sgRNAs targeting non-essential genomic regions (i.e. AAVS safe-harbor region) would be better controls when testing relative proliferation and H3K79me2.
5. To explore the function of the R2 element, several R2 variants of DOT1L were over-expressed in wild-type MLL-AF9 cells in which endogenous DOT1L is expressed. For a fair comparison, it is preferred to express exogenous wt and mutant DOT1L in cells without endogenous DOT1L. This can be achieved by designing exogenous wt and mutant DOT1L harboring synonymous mutations at the KMT core, followed by KO with an gRNA that has perfect match to the endogenous sequence but has mismatches to the exogenous sequences (ref. PMID 31586052).

minor comment:

1. In Britt Adamson's direct-capture Perturb-seq paper (PMID: 32231336), they mentioned that integration of CS1 at 3' end of sgRNA could compromise CRISPR activity, therefore they didn't recommend this design. In this manuscript, CS1 was inserted to the 3' end of sgRNA and is shown to work well according to RFP inactivation assay and sc-tiling data. Is there any explanation?
2. Some typos in the manuscript, for example in Figure S1, the primer should be CS1_R01.

We thank the reviewers for their thoughtful and constructive critiques of the manuscript. We have added new data/analyses to address comments, and we think the manuscript is significantly improved. We also modified the original “Introductory Paragraph” to become “Abstract” (revised page 2) and include an “Introduction” (revised page 2-3) to adhere to the Nature Communication format. Below is our point-by-point response (blue) to the reviewers’ comments.

REVIEWER COMMENTS

Reviewer #1 (Remarks to the Author):

General comment-summary:

Yang and colleagues combine CRISPR/Cas9-mediated editing with single cell RNA sequencing (that they call “sc-tiling”) to dissect the role of the H3K79 methyltransferase DOT1L in leukemic cells. Hereby they established a library of >600sgRNAs (3’ fused to a capture sequence) targeting the Dot1L ORF of 4.6kb, delivered by viral transduction into MLL-AF9 fusion gene immortalized mouse hematopoietic cells (Fig.1a-b). Sc-RNA sequencing revealed particular regions within the Dot1L ORF which resulted in differentiation or maintained an immature phenotype (Fig.1c-f). They found several regions in and outside the KMT domain that seemed essential to maintain the undifferentiated state (Fig.1g-h). They identified 3 critical regions in the N-terminal Dot1L ORF, including “R” between KMT and the AF9 binding domain (Fig.2). They also used the sc-tiling approach to identify regions that would modify cellular response to a well-characterized small molecule DOT1L-KMT inhibitor (EPZ5676).

Hereby they observed that while the N-terminal part of R (“R1”) enhanced, disrupting of the C-terminal part of R (“R2”) impaired inhibitor activity. Mutational analysis revealed several critical residues in R2, and structural modeling suggested a regulatory mechanism of KMT activity involving R1 and R2 (Fig.3a-f). Interestingly, they identified multiple single-nucleotide variants in patient samples in the R2 region of DOT1L which, when expressed in MLL-AF9 expressing cells increased their resistance towards EPZ5676 (Fig.3h-j). Overall, this is a well-written and interesting paper showing how the combination of CRISPR/Cas9 editing with scRNA sequencing can help to better understand the structure/functional relations of a particular pharmaceutical target and the strategy of inhibition. Many findings that emerge from this technically sophisticated approach seem however of confirmatory nature to the current knowledge except the identification of a particular stretch of the DOT1L ORF that when mutated provides resistance against the EPZ5676. This observation raises interesting questions about the biological relevance of such variants/alterations for cancer/leukemia patients, particularly those that showed limited response to small molecule DOT1L inhibitors in clinical trials. A better characterization of the variants in this particular region would further increase the impact of this paper beyond the high technical achievements.

Specific comments:

1. The identification of a region “R2” that when altered leads to resistance to small molecule DOT1L inhibitors seems novel and interesting. It would be important to provide more

information about these rare variants observed in 104/54510 “patient samples”. (1) What kind of samples were those? (2) From cancer patients or other diseases? (3) What kind of cancers? (4) Were the variants identified upon de novo diagnosis or after chemotherapy? (5) Can such variants also be acquired and/or selected during chemotherapy of human cancers? (6) Are these variants present in human cancer cell lines? (7) What about normal human beings: do we find DOT1L germline variants in this region?

- (1-2) The 54,510 "patient samples" included in this study were from the cBioPortal (<http://www.cbioportal.org>). This database initiated by Memorial Sloan Kettering Cancer Center (MSK) is now maintained by a multi-institutional team consisting of MSK, Dana Farber Cancer Institute, Princess Margaret Cancer Centre in Toronto, Children's Hospital of Philadelphia, The Hyve in the Netherlands, and Bilkent University in Ankara, Turkey. This comprehensive database collects cancer genomics information from diverse cancer types and multiple clinical studies.
- (3) To provide the detailed Study/Sample/Cancer Type information of the 19 missense variants observed in the R2 region, we have now provided a revised Supplementary Table 1 to include the interactive links to Study ID and Sample ID (including clinical data) for each variant discussed in this study.
- (4-5) In our study, the genomics information from all cancer types collected in cBioPortal was considered. Two [D610Y (small cell lung cancer) and E646K (breast cancer)] out of the 19 variants observed in the R2 region in this database have more than one sequencing datapoint. In both cases, the multiple sequencing results examined different metastasis tumors from the same patient. Therefore, our current analysis from the cBioPortal database does not provide sufficient information to examine whether these variants in the R2 region were associated (acquired/selected) with chemotherapy.
- (6) We examined the 1,457 human cell line sequencing information from the CCLE database (Cancer Cell Line Encyclopedia; BROAD Institute). A total of 6 variants were observed in the DOT1L R2 region (summarized in the table below; also see revised Supplementary Fig. 13a), including one missense variant G594S detected in RCCFG2 cells (clear cell renal carcinoma). The G594S variant was also observed in the cBioPortal dataset (adrenocortical carcinoma; kidney).

Cell Line	Tissue Type Classification	Variant Classification	Reference Allele	Tumor Seq Allele	Protein Change
DND41	HAEMATOPOIETIC_AND_LYMPHOID_TISSUE	Silent	G	A	L578L
MFE319	ENDOMETRIUM	Silent	C	T	L559L
KARPAS45	HAEMATOPOIETIC_AND_LYMPHOID_TISSUE	Silent	G	A	E639E
NCIH513	PLEURA	Silent	C	G	L592L
RCCFG2	KIDNEY	Missense	G	A	G594S
PECAPJ34CLONEC12	UPPER_AERODIGESTIVE_TRACT	In_Frame_Del	ACA	-	N562del

- (7) We examined the SNPs in the DOT1L R2 region from the Single Nucleotide Polymorphism Database (dbSNP; NCBI), including genomic information from 1000Genome, ExAC, TOPMed, GnomAD, GoESP. Eight out of 19 missense DOT1L variants discussed in our study were observed in the dbSNP Database as potential germline variants (summarized in the table below; also see revised Supplementary Fig. 13b).

Residue Variant	Variant ID	Location	Alleles	Evidence
A567V	rs1470399067	19:2213888	C/T	Frequency~gnomAD
R590C	rs371142941	19:2213956	C/T	Frequency~ESP~ExAC~TOPMed~gnomAD
A591T	rs776577348	19:2213959	G/A	Frequency~ExAC~TOPMed~gnomAD
A591V	rs376751030	19:2213960	C/T	Frequency~ESP~ExAC~TOPMed~gnomAD
R593C	rs200802307	19:2213965	C/T	Frequency~1000Genomes~ESP~ExAC~TOPMed~gnomAD
R593H	rs370589055	19:2213966	G/A	Frequency~ESP~ExAC~gnomAD
G594S	rs376032475	19:2213968	G/A	Frequency~ExAC~TOPMed~gnomAD
T613M	rs756254147	19:2214510	C/T	Frequency~ExAC~TOPMed~gnomAD

We would like to thank the reviewer's suggestion to investigate these clinically relevant genomics databases. The presence of these variants in normal and diseased human population supports the potential impact of this study on future personalized precision medicine. We have included these in the revised manuscript (page 9 line 9; see also revised Supplementary Fig. 13 and Supplementary Table 1)

2. Fig.1g needs to be better explained for non-experts. from far it appears to have more critical regions, particular in the KMT regions that reach a similar score, than the ones that are described in more detail in the structure in Fig.1h?

We appreciate the reviewer's suggestion to provide a more detailed labeling of the critical regions in Fig. 1g. In the revised Fig. 1g (comparison attached below), we highlighted four functionally essential regions in the KMT core that were discussed in Fig. 1h. These include (1) P133-T139 D1 loop, (2) SAM pocket (key residues shown in Supplementary Fig 6a), (3) R282 loop, and (4) T320-K330 helix.

3. Fig.2a compare Dot1L ORF regions with impact on gene expression with the “AF9-binding motif” showing “a moderate correlation. However, in the experiments showed in Fig.3a, this region seemed very sensitive in control as well as EPZ5676 treated cells, in fact reaching the lowest NCS? The authors may clarify that in the text.

DOT1L has two reported functions in MLL-r leukemia through (1) KMT core's H3K79 methyltransferase activity to maintain open-accessible chromatin (Ref #18), and (2) AF9-binding motif to recruit the AF9-containing “Super Elongation Complex” (SEC; consist ENL, ELL, AF9, and AF4) to support gene transcription (Ref #33,34). While the sgRNAs targeting the AF9-binding motif exerted a similar cell killing effect as targeting the KMT core, sc-Tiling (a transcriptomic-based single-cell CRISPR profiling) observed detectable differences between these two functional elements with a modulate correlation score (Pearson correlation $\cong 0.75$). Conversely, sgRNAs targeting the R1-element exerted similar epigenetic (loss of H3K79me2) and transcriptomic (Pearson correlation > 0.8) profiles as targeting the KMT core, suggesting the collaborative nature between the R1-element and KMT core. The fact that

cell killing through targeting the AF9-binding motif did not impair the H3K79me2 level (revised Fig. 2e) supports the utility of sc-Tiling to provide a superior characterization of the functional elements than the traditional survival-based CRISPR gene scan. We appreciate the reviewer's suggestion and have included these notions in the revised manuscript (page 8 lines 10-20).

Reviewer #2 (Remarks to the Author):

In this manuscript, Lu et al integrated a CRISPR-tiling screen with the direct-capture Perturb-seq, named 'sc-tiling', to identify gene functions at a sub-gene resolution and investigate the regulatory mechanisms within gene-coding regions. Using this new approach, they discovered a novel functional R domain in DOT1L that coordinated with KMT domain to regulate histone modification. It is interesting that the R1 and R2 elements in the R domain exhibit opposite responses during DOT1L-inhibitory treatment, indicating that this self-regulatory R domain dictates the response to DOT1L-targeted therapy.

Overall, this is a well-organized study with attractive novel points. This is the first report of an application of tiling screen at a single cell resolution. The discovery of the opposite response of the R1 and R2 is interesting too. Meanwhile, I have the following questions and suggestions that might be helpful for the refinement of the manuscript.

1. What are the pros and cons of sc-tiling compared to the survival-based pooled tiling screens? This is a critical question that needs to be addressed. A comparison of the sc-tiling result in Fig. 2A and the survival screen in Fig. 2B can be useful. I'd suggest performing a PCA analysis on the correlation matrix of Fig. 2A and to compare the principal component to the viability profile in Fig. 2B. This will tell us which amino acids are functionally associated with the KMT core at the transcriptome level, in comparison to those at the cell phenotypical level. More discussion on the pros and cons of the two alternative methods is preferred too.

We want to thank the excellent suggestions from the reviewer. To our knowledge, DOT1L has two reported functions in MLL-r leukemia through (1) KMT core's H3K79 methyltransferase activity to maintain open-accessible chromatin (Ref #18), and (2) AF9-binding motif to recruit the AF9-containing "Super Elongation Complex" (SEC; consist ENL, ELL, AF9, and AF4) to support gene transcription (Ref #33,34). The fact that cell killing through targeting the AF9-binding motif does not impair the H3K79me2 level (revised Fig. 2e) argues the catalytic-independent role of the AF9-binding motif in DOT1L.

Following the reviewer's suggestion, we performed the PCA analysis of the sc-Tiling (data from Fig. 2a) and cross-compared the PC1 score with the survival-based CRISPR scan score (day 12 shown in Fig. 2b) of individual amino acid residues. As shown below (see also revised Fig. 2c), the distribution of KMT core residues (black) overlap with the AF9-binding motif residues (green) when evaluated by the survival CRISPR scan score (x-axis). In contrast, the sc-Tiling PC1 score (y-axis) can distinguish the AF9-binding motif from the KMT core. This combination analysis suggested by the reviewer also revealed the critical amino acid residues located in the center of R1 (E489 – L515; red) that functionally

associated with the KMT core at both the transcriptomic level and cellular survival phenotypes, further improved the resolution of the sc-Tiling analysis. Of note, all three sgRNAs for R1 element used in this study (target amino acids 495, 505, and 510; also see Supplementary Fig. 7b) target within the “R1 center”. We have updated these notions in the revised manuscript (revised Fig. 2c; page 5 line 26 to page 6 line 5; page 8 lines 10-16).

Based on these new analyses, we envision one major advantage of sc-Tiling is the ability to distinguish underlying mechanisms between the functional elements within a protein. This could be significant as the traditional survival CRISPR scan can only address the overall cellular fitness with limited power to imply the underlying mechanisms of a particular domain. Furthermore, the transcriptomic profiling in sc-Tiling argues the potential of this approach in dissecting functional elements that participate in diverse cellular processes (e.g., metabolism, cell fate decision, tissue homeostasis, etc.) that the end phenotypes might not be the cellular survival or proliferation. We updated these notions in the revised manuscript (page 8 lines 16-20).

2. In the sc-tiling screen, samples were collected at day 3 after transduction. Why this timepoint is chosen? Is this because DOT1L is very essential in MLL-AF9 cells? I wonder if day 3 is sufficient for some relatively inefficient sgRNAs to achieve genome editing. The authors should provide explanation and address the potential limitation.

We agree with the reviewer’s comment that the time point of the sc-Tiling is critical. In our preliminary studies, we monitored the cell viability and the expression level of Hoxa9 (a DOT1L driven gene) in MLL-AF9 cells transduced with a sg-KMT (below; see also revised Supplementary Fig. 2d,e). In these assays, we constantly observed a significant reduction of cell viability upon 6 days of culture (this phenomenon was also observed in the survival CRISPR scan shown in Fig. 2b, i.e., first sgRNA dropout phenotype observed on day 6). We also noted that the expression of Hoxa9 (which serves as a DOT1L functional reporter) was significantly attenuated by day 3, whereas the cells remained largely viable. To avoid the

influence of “cell death signature” dominating the transcriptomic analysis, we selected day 3 to perform the sc-Tiling screen. We updated these notions in the revised Supplementary Fig. 2d,e.

We echo the reviewer’s concern regarding the genome editing efficiency in CRISPR screens. As it remains technically challenging to investigate the cutting efficiency of all ~600 sgRNAs in the library, we focused on examining the editing efficiency of the 14 DOT1L sgRNAs validated in this study. We collected the genomic DNA from day 3 transduced cells, PCR amplified the CRISPR targeted genomic regions, and submitted for Illumina PE150 sequencing to determine the mutation % of each CRISPR targeted locus (below; see also revised Supplementary Fig. 7c). Our results showed an average of $94.8 \pm 3.4\%$ editing efficiency, with 13 out of 14 tested DOT1L sgRNAs exerted higher than 90% editing efficiency by day 3. However, one sgRNA targeting AF9-binding motif (sg_mDot1l_2645_s) achieved only ~85% editing.

We acknowledge the variable cutting efficiency, potential for off-targeting, and the mosaic effect [i.e., generation of random mutations]) of individual sgRNA remain concerns in the CRISPR sc-Tiling. We propose that we could increase the statistical confidence and minimize the noise associated with individual sgRNAs by considering multiple sgRNAs via a local-smoothing strategy. We included these notions in the revised manuscript (page 8 lines 21-25; revised Supplementary Fig. 7c).

3. As shown in Fig. 2D, the H3K79me2 level is decreased in sg-KMT and sg-R1 cells without DOT1L-inhibitory treatment. Does the H3K79me2 level alter upon sg-R2 or Q584P (Fig. 3I) without DOT1L-inhibitory treatment?

While sg-KMT and sg-R1 reduced the H3K79me2 level in the targeted cells, transduction of sg-R2 did not significantly impact the overall H3K79me2 level (below; see also revised Supplementary Fig. 9b). We envision that only a fraction of the random mutations created by sg-R2 behaves like the EPZ5676-resistant variants.

The relative H3K79me2 level in Fig. 3i without DOT1L-inhibitor were (WT) 1.000, (Q584P) 1.416, (L626P) 1.302, and (C637G) 2.069 (now available in Source Data). We also validated the increased H3K79me2 by these three human DOT1L variants in MLL-AF9 cells transduced with a sgRNA diminishing the endogenous mouse DOT1L's activity (Supplementary Fig. 11c; more details in the response to question #5).

4. In Figures 2C and 2D, sg-Luc was used as the control. This doesn't introduce cleavage to the DNA. The sgRNAs targeting non-essential genomic regions (i.e. AAVS safe-harbor region) would be better controls when testing relative proliferation and H3K79me2.

We examined the effect of sg-AAVS (GGGGCCACTAGGGACAGGAT) on MLL-AF9 leukemic cells and observed similar proliferation and H3K79me2 level as compared to sg-Luc (below). We updated this additional control in the revised Supplementary Fig. 9a,b to support our original finding.

5. To explore the function of the R2 element, several R2 variants of DOT1L were over-expressed in wild-type MLL-AF9 cells in which endogenous DOT1L is expressed. For a fair comparison, it is preferred to express exogenous wt and mutant DOT1L in cells without endogenous DOT1L. This

can be achieved by designing exogenous wt and mutant DOT1L harboring synonymous mutations at the KMT core, followed by KO with an gRNA that has perfect match to the endogenous sequence but has mismatches to the exogenous sequences (ref. PMID 31586052).

Since the 19 missense DOT1L-R2 variants in the cBioPortal database exhibited allele frequency between 0.06 and 0.58 (median 0.29; revised Supplementary Table 1), it is predicted that most of these variants existed as heterozygous in patients. Therefore, we examined the function of exogenous wild-type and mutant human DOT1L in the MLL-AF9 cells while the endogenous mouse DOT1L remains active (data presented in Fig. 3h and 3i).

We also agree with the reviewer’s suggestion that further characterization of the drug-resistant DOT1L variants in the MLL-AF9 cells missing an endogenous DOT1L activity will strengthen the conclusion. To select a sgRNA that targets the endogenous mouse DOT1L activity but not affecting the exogenous human DOT1L cDNA constructs, we first evaluated the amino acids and coding DNA sequences of the human vs. mouse DOT1L within the KMT domain (M1-P332). We found that while the amino acid sequences are highly conserved (330/332 = 99.3% identical), the coding DNA sequences are more degenerated (891/996 = 89.5% identical) between the human vs. mouse DOT1L (below; also see revised Supplementary Fig. 11a,b). We also found that one of the KMT core sgRNAs used in our study (sg-mDot1l-551-s) only targets the endogenous mouse DOT1L (human DOT1L cDNA contains 2 nucleotide mismatches near the 3’ end of sgRNA, plus 1 alternation in the PAM sequence [i.e., human cDNA does not have this PAM]).

Human vs. Mouse DOT1L KMT domain
(330/332 = 99.3% identical)

Hu 1	MGEKLELRKSPVGAEPVAVYPWPLPVYDKHDAHEIETIRWCEEIPDLKLAMENYVL	60
Ms 1	60
Hu 61	IDYDTKSFESMQRKCDKYNRAIDSIHQWKGTTQPMKLNTRPSTGLLRHILQQVYVNSHT	120
Ms 61	120
Hu 121	DPEKLNMYEPFSPVEVGETSFDLVAQIDEIKMTDDDLFVDLGSVGVQVVLQVAAATNCK	180
Ms 121	180
Hu 181	HHYGVKADIPAKYAETMDREFRKMWKYGGKHAEYTLERGDFLSEWRERIANTSVIFV	240
Ms 181	240
Hu 241	NNFAGPEVDHQLKERFANMKEGGRIVSSKPFAPLNFIRNSRNLSDIGTIMRVVELSPLK	300
Ms 241	300
Hu 301	GSVSWTGKPVSYLHTIDRTILENYFSSLKNP	332
Ms 301	332

Human vs. Mouse DOT1L KMT cDNA
(891/996 = 89.5% identical)

Hu 1	ATGGGGGAGAAGCTGGAGCTGAGACTGAAGTCGCCGTGGGGCTGAGCCCGCCGTCTAC	60
Ms 1	60
Hu 61	CCGTGGCCGCTGCCGTCTACGATAAACATCACGATGCTGCTCATGAAATCATCGAGACC	120
Ms 61	120
Hu 121	ATCCGATGGGCTGTGAAGAAATCCCAGATCTCAAGCTCGTATGGAGAATTACGTTTTTA	180
Ms 121	180
Hu 181	ATTGACTATGACACCAAAGCTTCGAGAGCATGCAGAGGCTCGCGACAAGTACAACCGT	240
Ms 181	240
Hu 241	GCATCGACAGCATCCACAGCTGTGGAAAGGCCACACGAGCCATGAAGCTGAACACG	300
Ms 241	300
Hu 301	CGCCCGTCCACTGGACTCTGCGCCATATCCTGCAGCAGGTCTACAACACTCGGTGACC	360
Ms 301	360
Hu 361	GACCCCGAGAAGCTCAACAACTACGAGCCCTTCTCCCGAGGTGTACGGGGAGACCTCC	420
Ms 361	420
Hu 421	TTGACACTGGTGGCCAGATGATTGATGAGATCAAGATGACCGACGACGACCTGTTTGTG	480
Ms 421	480
Hu 481	GACTTGGGAGCGGTGTGGCCAGGTCGTGCTCCAGTTGTGCTGCCACAACTGCAAAA	540
Ms 481	540
Hu 541	CATCACTATGGCTCGAGAAAGCAGACATCCCGGCCAAGTATGCGGAGACATGGACCGG	600
Ms 541	600
Hu 601	GAGTTCAGGAAGTGGATGAAATGGTATGGAAAAAAGCATGCAGAATACACATTGGAGAGA	660
Ms 601	660
Hu 661	GCGATTCTCTCAGAAAGAGTGGAGGGAGCGAATCGCCAACACGAGTGTATATTTGTG	720
Ms 661	720
Hu 721	AATAATTTTGCCTTTGGTCTGAGGTGGATCACCAGCTGAAGGAGCGGTTTTCGAAACATG	780
Ms 721	780
Hu 781	AAGGAAGGTGGCAGAATCGTGCTCGAAACCTTTGCACCTCTGAACTTCAGATAAAC	840
Ms 781	840
Hu 841	AGTAGAAACTTGAGTACATCGCCACCATCATGCGCGTGGTGGAGCTCTGCCCTTGAAG	900
Ms 841	900
Hu 901	GGCTCGTGTCTGTCGACGGGGAAGCAGTCTCTACTACTGCACACTATCGACCGCACC	960
Ms 901	960
Hu 961	ATACTTGAAAATAATTTTCTAGTCTGAAAAACCA	996
Ms 961	996

We then transduced this sg-KMT core into the MLL-AF9-Cas9⁺ cells expressing the YFP-tagged human DOT1L cDNA (WT, Q584P, L626P, C637G), and examined the H3K79me2 level and EPZ5676 response (below; also see revised Supplementary Fig. 11c,d). Similar to the results shown in Fig. 3h and 3i, these drug-resistant DOT1L variants exhibited increased H3K79me2 and resistant index to EPZ5676 in MLL-AF9 cells with inactivated endogenous DOT1L. We updated these additional tests in the revised Supplementary Fig. 11 to support our original finding.

minor comment:

1. In Britt Adamson’s direct-capture Perturb-seq paper (PMID: 32231336), they mentioned that integration of CS1 at 3’ end of sgRNA could compromise CRISPR activity, therefore they didn’t recommend this design. In this manuscript, CS1 was inserted to the 3’ end of sgRNA and is shown to work well according to RFP inactivation assay and sc-tiling data. Is there any explanation?

When we initiated the sc-Tiling project in 2019, we consulted the 10X Genomics and received two recommendations for the CS1 insertion into our sgRNA vector (1) integration in sgRNA hairpin and (2) integration in sgRNA 3’ end. Both approaches have been demonstrated compatible with the 10X Genomics GEM v3.1 for CS1 capturing (10X Genomics information shown below). We selected the 3’ end insertion to modify our sgRNA vector and observed comparable RFP cutting efficiency as our original sgRNA backbone (edit ~95% cells; Supplementary Fig. 1b).

Information provided by 10X Genomics

We acknowledge the direct-capture Perturb-seq published by Adamson's lab (Replogle et al. 2020 Nature Biotech; PMID: 32231336; Ref#11) that revealed a superior cutting performance by integrating the capture sequences in the stem-loop 2 of sgRNA. One possible explanation for the sufficient cutting detected in our 3' CS1 sgRNA backbone system is the selection of monoclonal MLL-AF9-Cas9⁺ cells that exerted the strongest CRISPR activity for the screens (Supplementary Fig. 2c). We speculate that the adequate Cas9 activity in our cell model could render robust editing while the sgRNA/CS1 backbone designs could be further optimized.

2. Some typos in the manuscript, for example in Figure S1, the primer should be CS1_R01.

We have corrected this error as recommended.

Reviewers' Comments:

Reviewer #1:

Remarks to the Author:

The authors nicely responded to all the points raised by both reviewers which clearly improved the overall quality of this MS further.

Reviewer #2:

Remarks to the Author:

I appreciate the time and efforts of the authors to revise the manuscript. There are plenty of new data to support the authors' arguments in the reply. All my questions have been carefully addressed, and I have no further questions.